# Predictors of early newborn deaths at Dar es Salaam public regional referral hospitals: A prospective observational hospital-based study

Grace Frank Mhando[1,2,3☉], Salvatory Florence Kalabamu[1], Maulidi Rashidi Fataki[1], Christina Clinton Galabawa[1,3,4], Kelvin Melkizedeck Leshabari [3,5,6☉] *

1 Dept of Paediatrics/Child Health, Faculty of Medicine, Hubert Kairuki Memorial University, Dar es Salaam, Tanzania, 2 Dept of Paediatrics & Child Health, Amana Regional Referral Hospital, Dar es Salaam, Tanzania, 3 NEOnatal Network (NEON-r) research group, Registered Trustees of Ultimate Family Healthcare, Dar es Salaam, Tanzania, 4 Port Health Centre, Tanzania Ports Authority, Dar es Salaam, Tanzania, 5 Ageing Initiative in sub-Saharan Africa (AISA) research group, Registered Trustees of Ultimate Family Healthcare, Dar es Salaam, Tanzania, 6 H3 Clinical Research Unit, I-Katch Technology Ltd, Dar es Salaam, Tanzania

☉ These authors contributed equally to this work.
* celsius_lx@yahoo.co.uk, kelvin.leshabari@ufht.or.tz

## Abstract

### Introduction

Newborn deaths are still a concern to global health systems. Increased rates of documented hospital-based births are incongruent to newborn survival chances worldwide. Factors for that rather paradoxical observation are largely unknown. We aimed to assess predictors of early newborn deaths at representative metropolitan referral health facilities in Africa.

### Materials & methods

We designed a prospective, analytical, hospital-based study in neonatal units at Dar es Salaam public regional referral hospitals, Tanzania. Neonates who died within the first 7 days of life were the target population. A pre-designed Case Report Form was the main tool for data collection. Multivariable binary logistic regression model was fitted to account for predictors of early newborn deaths after appropriate linear model validation. Proportion of early newborn deaths was the outcome variable.

### Results

We recruited and analysed 2212 neonate-days of follow-up. Prevalence of early newborn deaths was 28.1%. Birth asphyxia ($\chi^2 = 20.4$, df = 1), preterm delivery ($\chi^2 = 5.36$, df = 1) and respiratory distress syndrome ($\chi^2 = 30.94$, df = 1) were associated with early neonatal outcomes. Predictors of early newborn deaths were neonatal respiratory

**Data availability statement:** All relevant data are within the paper and its Supporting information files.

**Funding:** The author(s) received no specific funding for this work.

**Competing interests:** The authors have declared that no competing interests exist.

rate (Tachypnoea – A.O.R.: 2.28 (95% CI.: 1.44–5.79); Bradypnoea – A.O.R.: 1.9 (95% C.I.: 1.02–12.3) and gestational age (Preterm delivery – A.O.R.: 1.48, 95% CI.: 1.11–2.09 and Post-term delivery – A.O.R.: 5.05, 95% C.I.: 4.49–32.0).

### Conclusions

Early newborn deaths rate was relatively high in this study population. Newborns' respiratory rates and gestational age at delivery were significant clinical factors associated with early newborn deaths in this study.

### Introduction

Early newborn (neonatal) deaths remain a major public health concern worldwide, especially in Africa [1–3]. Early newborn deaths refer to mortal events happening within the first seven days of life, and late newborn deaths are those that happen after seven days but within the first 28 days of extra-uterine life [4,5]. At present, the demographic data on neonatal death risk factors tend to be controversial. For instance, the three commonest factors associated with early neonatal deaths in developed countries are prematurity, congenital abnormalities, and low APGAR scores [6]. In contrast, regions like Africa south of Saharan desert report the greatest rate of early neonatal deaths, mainly due to neonatal infections (e.g., sepsis, pneumonia), birth asphyxia as well as complications associated with preterm delivery [7]. Dar es Salaam city is among the fastest growing cities in Africa [8]. Dar es Salaam is scheduled to be the third most populous city in Africa by 2050 [8]. The interplay for the factors associated with early neonatal deaths in developing countries (Tanzania inclusive) tend to be multifactorial and non-specific. At the very least, little is known about the dynamics and interplay of those factors early in life.

There are several factors published in literature that are associated with early neonatal deaths in the health systems [1,9,10]. Conveniently, they can be summarized under neonatal, maternal and hospital system factors. However, in some reported statistics, these factors are either contradictory or antagonistic, dependent on a number of mechanisms. For instance, whereas early neonatal conditions/infections/deaths are justifiably attributable to maternal/obstetric factors, it is common knowledge that in most cases, late neonatal infections are attributable to neonatal/hospital-related factors. However, little is known about the changes brought by current epidemiological and demographic transition, especially in African countries.

At present, we acknowledge death statistics to be important for demographic studies, clinical as well as public health analyses. Vital system statistics are still in their infancy stage in Tanzania and Africa in general. They are generally deficient in terms of their significant under-registration of deaths, especially among <5-year population; extensive misreporting of age as well as significant 'non-reporting' of age-specific deaths, notably those that occur outside hospital premises. Amidst all the known challenges, there is still a glimmer of hope in utilization of hospital statistics, especially for early newborn deaths if prospectively collected.

In East African zone, Tanzania bears the greatest burden of neonatal deaths, accounting for 40% of all deaths in children under the age of five [11]. The most consequential aspect being 65.5% of all reported deaths occurred within first 24 hours of life [11]. Besides, there is palpable evidence that Dar es Salaam city is among cities with highest neonatal deaths in Tanzania [12]. However, the dynamics to which early neonatal deaths contributes to overall neonatal deaths statistics as well as causal pathways associated with the early deaths in that specific population cohort remains speculative [3]. We hypothesized the same mechanisms to be also applicable to other African countries but never fully explored before.

In Tanzania the crude early neonatal death rate is about 18 per 1000 live births [3]. The statistic is based on a national survey report of 2019 [3]. The published survey report also revealed significant variations in neonatal deaths across regions, with some regional variations also displaying inherent within-regional differences in mortality figures [3]. Recent study has shown that there is an increase of newborn deaths in urban areas such as Dar es Salaam compared to those delivered in rural areas [12]. However, factors that caused (are associated with) that rather paradoxical findings were not explored. Habitually, most of the retrievable peer reviewed publications in this specific agenda had either cross-sectional or retrospective designs [3,6,11]. We could not rely on those statistical estimates given inherent biases associated with their reported study designs. We therefore conceptualized this prospective analytical hospital-based clinical research to assess early neonatal mortality rate and associated factors at typical African urban regional referral hospitals.

## Methods

### Definition of key terms

**Early neonatal period:** The timespan between live-birth and 7 days post-delivery.
**Early neonatal death/mortality:** death events among live-newborns within 7 days of extra-uterine life.
**Incident neonatal mortality rate:** proportion of newborn deaths within the first 7 days of life

### Study design and follow-up

We designed a prospective observational hospital-based clinical research study. Newborns were clinically followed-up immediately from birth up to (and including) day seven of extra-uterine life or death, which ever came first.

### Study settings

The study was conducted at neonatal units of Amana, Mwananyamala and Temeke regional referral hospitals in Dar es Salaam. The named hospitals serve as the only highest points of referral hospitals for the public within the five administrative municipalities in Dar es Salaam metropolitan. Specifically, Amana is the top public regional referral hospital for Ilala municipality. Mwananyamala serves as the highest regional referral health facility for both Kinondoni and Ubungo municipalities. Temeke is reserved as the highest point of public referral centre for Temeke and Kigamboni municipalities. Thus, for all practical purposes, we considered those referral hospitals as being representative to Dar es Salaam mega city's regional hospitals system.

### Study population

Newborns admitted at regional referral hospitals in Dar es Salaam city.

### Target population

All newborns who died within 7 days of life.

## Sampling procedure

All newborns born at Amana, Mwananyamala and Temeke between 17 May and 16 November 2023 were eligible to be included into the study. A minimum sample of at least 319 newborns was enough to achieve study power of at least 80% within 5% level of significance. Initial plan was to employ a simple random sampling (without replacement) technique but since the study resources allowed inclusion of all neonates who met the inclusion criteria in the study timespan, all neonates who met the study's inclusion (and without exclusion) criteria; and were delivered at the study settings during the study period were therefore recruited.

## Recruitment and study duration

Newborns were recruited immediately after birth at the study sites. Recruitment was from 17th May 2023 to (and including) 16th November 2023. Incident mortality rate was the main outcome measure for this specific agenda. Specifically, neonates born (and admitted) at study sites within 7 days of life were recruited into the study. Moreover, mothers of newborns at the study settings who were referred to higher facilities (i.e., Muhimbili National Hospital) had newborns retained at the facilities as per existing guidelines at Dar es Salaam regional referral hospitals; and therefore their newborn babies were also eligible to be part of the study. Neonates who were beyond seven days of life were excluded from this study.

## Data collection procedure

The pre-designed Case Report Form (CRF) was used for data collection. Specifically, it contained five different sections namely – *demographics*, *maternal history in ante-partum and labour times*, *data during birth* as well as *immediate and underlying causes of deaths*. An additional section on *hospital-related data* was also part of the CRF. The Case Report Form (CRF) was constructed by reflecting the study objectives and current good clinical practice guidelines. Before actual collection, the CRF was pilot-tested at two different private facilities for a duration of 5–7 days in order to assess measure of test-retest reliability as well as construct validity scores of tool items. Data collected also included verifications from RCH-cards, neonatal registration system (TEHAMA) as well as separate CRF-led interviews with mothers from each neonate admitted. Besides, each recruited neonate had physical examination performed by either the Principal Investigator (PI) or the research team members (MD-registrars at paediatrics depts. of Amana, Mwananyamala and Temeke hospitals). There was a 1-day research training for all research team members during pilot testing of the tool. Specifically, the training involved all research team members on aspects of good clinical and research ethics practice.

## Variable design and measurements

Early (≤ 7 days) neonatal death/mortality was the main outcome variable while a series of neonatal (age – in days, gender, birth weight – in kg, temperature in Celsius scale, heart rate – in beats/min, respiratory rate – breaths/minute, heart murmur, neurological state, abdominal distension, enlarged organ, abdominal masses, Apgar scores [1st and 5th minutes] at birth, oxygen saturation at birth, random blood glucose (in mmol/L) at birth, time [approximated to nearest hours] to start breast feeding, presenting complaint(s) in the neonatal wards; maternal (date of birth of the mother, highest level of education, mother's current occupation, religious affiliations, residence, marital status, mode of delivery, birth status [singleton or multiple gestation], preceding birth interval, number of antenatal visit during pregnancy, partographic findings, type of delivery [spontaneous vaginal delivery, Caesarean birth or instrumental delivery], history of maternal fever, meconium stained, premature rupture of membrane, chorioamniotis, antepartum haemorrhage) as well as hospital system factors [staff cadre level delivering the baby, time-to-entry into the neonatal ward] were set as independent variables.

## Data processing and analysis plan

All data collected were double entered in a pre-designed template using Epi info statistical software version 7.4 (Epi-Info ™ CDC – Atlanta, USA). Data cleaning (e.g., checking for missing value, errors in data entry) was done on EPI INFO software version 7.4 and then exported to SPSS (IBM – Illinois, USA) version 23 for analysis. Since all continuous data displayed skewed distributions, they were summarized using median and inter-quartile range while categorical data were summarized using frequency and proportion. Besides, for selected categorical variables, their association with early neonatal mortality were assessed using likelihood ratio-based chi-square test with corresponding degrees of freedom. On death patterns, the International Classification of Disease number eleven ICD-11 were applied to label underlying causes of deaths. Incident early neonatal death/mortality rate was computed by direct method. Simply stated, we considered the ratio between number of newborn deaths occurring within 7 days of extra-uterine life to newborn-time pair of follow-up. Moreover, we also included analysis findings of prevalence mortality rate among study participants. To analyze predictors for early deaths, a multivariable binary logistic regression model was used to analyze predictors for early neonatal mortality after appropriate validation of linear model assumptions. Besides, a test for interactions (effect modifiers) was also done during multivariable binary logistic regression model fitting.

## Ethical clearance and informed consent

Ethical clearance for the study was sought from the Institutional Research Committee of the Hubert Kairuki Memorial University in Dar es Salaam, Tanzania. Permission to conduct the study at the study sites was sought from Medical Officer in-charge of Amana, Mwananyamala and Temeke regional referral hospitals. Prior to recruitment of each neonate, mothers were approached for written informed consent that included brief summary of the study goals, risks and benefits of their newborns participations, voluntary nature of each baby participation (including the clause that inclusion/withdrawal to participate was free at any time without any compromise to available care at the facility), and that their babies were to be subjected to routine physical examination on enrollment.

## Results

We successfully recruited and analysed a total of 2212 neonates-days of follow-up at Amana, Mwananyamala and Temeke regional referral hospitals in Dar es Salaam. We observed a slight male preponderance (n = 200, 54.1%) among followed-up newborns delivered at study sites. Moreover, majority (n = 263, 71.1%) were delivered by spontaneous vertex delivery, Caesarean delivered babies accounting for 28.4% (n = 105) and the rest (0.5%) being delivered by assisted deliveries. On maternal baseline clinical characteristics of studied newborns – more than half (50.8%) were multigravidae, 1.1% had obstructed labour, antepartum haemorrhage was rarely (0.3%) observed. On average, mothers of newborns attended a median of 4 (IQR: 3–5) antenatal care clinics. Moreover, on socio-demographic baseline characteristics included more than half (61.8%) had secondary education, 48.4% were self-employed while 43.2% were unemployed. Besides, we also found the prevalence of deaths among followed-up babies to be 28.1% at the study sites during the study period. Interestingly, more than half (52.8%) of the reported early neonatal deaths occurred within the first 24 hours after delivery. Table 1 below summarises the baseline characteristics of studied neonates:

In total, the incident unit of *neonate-days* of follow-up was derived by following up 370 newborns between May and November 2023. Besides, it was also of interest to assess association between neonatal diagnoses (on admission) and early neonatal death. The decision to select diagnoses reflected top 10 diagnoses reported in neonatal wards at Dar es Salaam hospitals. Table 2 below highlights the summary statistics:

Since all the analysed variables above (Table 2) were categorical by design, the chi-square test (with corresponding degrees of freedom) adopted Likelihood Ratio Chi-square test statistics. Besides, the fact that observed statistics between variables had unequal frequencies, all Chi-square test scores were subjected to Yates' continuity correction. Lastly, the

**Table 1. Selected baseline characteristics of newborns followed up at Amana, Mwananyamala & Temeke regional referral hospitals (May – November 2023).**

| Selected Neonatal | Continuous | Variables |
|---|---|---|
| **Continuous variable** | **Median*** | **Inter-quartile range*** |
| Newborn age (in completed days) | 6 | 4 - 7 |
| Apgar score at 1st minute | 7 | 6–8 |
| Apgar score at 5th minute | 10 | 7–10 |
| Birth weight (kg) | 2.7 | 2.0–3.2 |
| Gestational age at birth (weeks) | 38 | 35–39 |
| Oxygen saturation on admission (%) | 90 | 87 - 95 |
| Respiratory rate (breaths per minute) | 60 | 49–68 |
| Temperature - ºC | 36.8 | 36.6–37.8 |

**Note:** *Median and inter-quartile range were used as summary measure of central tendency for all continuous variables since they all had skewed distribution during initial data exploration.

**Table 2. Selected neonatal diagnoses on admission associated with early neonatal outcomes.**

| Factors | Alive | Dead | Chi-square** | df | p-value |
|---|---|---|---|---|---|
| **Birth asphyxia** | | | | | |
| Evident | 75 | 56 | 20.4 | 1 | 0.000 |
| Non-evident | 191 | 48 | | | |
| **Prematurity** | | | | | |
| Evident | 82 | 46 | 5.36 | 1 | 0.02 |
| Non-evident | 184 | 58 | | | |
| **Early-onset neonatal sepsis** | | | | | |
| Evident | 110 | 32 | 3.108 | 1 | 0.078 |
| Non-evident | 156 | 156 | | | |
| **Respiratory distress (RDS)** | | | | | |
| Evident | 15 | 28 | 30.94 | 1 | 0.000 |
| Non-evident | 251 | 76 | | | |
| **Meconium aspiration** | | | | | |
| Evident | 17 | 13 | 2.97 | 1 | 0.085 |
| Non-evident | 249 | 91 | | | |

**\*NB:** Total was greater than (N = 370) due to multiple diagnoses on admission. For instance, some neonates were admitted with both birth asphyxia and prematurity. ** Chi-square test statistics included Yates' continuity corrections.

main analysis included estimates of both crude and multivariable binary logistic regression for factors linearly associated with early newborn deaths at study settings. Prior to fitting a binary logistic regression model, we assessed the residuals of all maternal, neonatal and hospital related factors for *normality*, *homoscedasticity*, *autocorrelation*, significant *multicollinearity* as well as *linearity* assumptions against the outcome variable. For all categorical variables had dummies made and the reference category/values as indicated.

Majority of the studied factors failed to meet the assumptions of linear model fitting in exception to maternal age and gestational age [in completed weeks] at delivery (maternal factors); respiratory rate, Apgar scores (at both 1st and 5th minutes), temperature, gender and birth weight (neonatal factors). For hospital system characteristics, we included cadre

of the healthcare worker who delivered the baby, hospital name (coded as centre ID), number of qualified neonatologists, availability of neonatal ICU as well as presence/absence of visible emergency neonatal resuscitation protocol. In exception to hospital name (centre ID), all other hospital-related factors failed in one or more tests of linear model assumptions during validation of linear model fitting. They were thus discarded.

Table 3 below provides the statistical summaries of interest from the combined univariate and multivariable binary logistic regression outputs:

## Discussion

Our analysis revealed almost a third of all sick admitted babies in the study settings died during follow-up period. This observed statistic is interesting for the following reasons. First, there appears to be missing targets in most interventions and strategic efforts to reduce newborn deaths at the study settings. Second, as per the final multivariable analysis, factors associated with early newborn deaths among Dar es Salaam hospitals were unlikely to be uniform. For instance, whereas babies delivered at Amana regional referral hospital were on average up to 60% less likely to die during the first seven days of life compared to those born at Temeke regional referral hospital, babies born at Mwananyamala regional referral hospital were observed to have the highest statistical risk of deaths of all the three studied hospitals. Since the

**Table 3. Multivariable analysis of factors associated with early newborn deaths in Dar es Salaam public regional referral hospitals.**

| Variable name | Crude/unadjusted O.R. | Analysis 95% C.I. | P-value | Adjusted/ A.O.R. | Multivariable 95% C.I. | analysis P-value |
|---|---|---|---|---|---|---|
| **Constant** | **0.071** | **0.04–0.09** | **0.000** | **0.066** | **0.04–0.341** | **0.000** |
| | | MATERNAL | | | FACTORS | |
| Mode of delivery – svd | 0.736 | 0.44–1.24 | 0.25 | 1.039 | 0.57–1.91 | 0.901 |
| Mode of delivery – C/S | 1 | | | 1 | | |
| ANC visits (frequency) | 0.882 | 0.72–0.93 | 0.003 | 0.882 | 0.75–1.03 | 0.118 |
| Maternal age –<18 y | 1.313 | 0.67–2.47 | 0.445 | 1.74 | 0.88–2.92 | 0.07 |
| Maternal age ->35 y | 1.22 | 0.61–2.67 | 0.598 | 1.08 | 0.75–3.29 | 0.838 |
| Maternal age – 18–35 y | 1 | | | 1 | | |
| | | NEONATAL | | | FACTORS | |
| **Tachypnea (≥ 60b/min)** | **3.17** | **1.91–5.2** | **0.000** | **2.28** | **1.44–5.79** | **0.02** |
| **Bradypnoea (<40b/min)** | **3.68** | **1.47–17.7** | **0.04** | **1.9** | **1.02–12.3** | **0.03** |
| **Eupnoea (40–59 b/min)** | **1** | | | **1** | | |
| Temperature (oC) | 0.65 | 0.44–0.76 | 0.001 | 0.57 | 0.23–1.89 | 0.296 |
| 1st minute Apgar score | 0.804 | 0.72–0.90 | 0.000 | 0.889 | 0.56–1.37 | 0.593 |
| 5th minute Apgar score | 0.837 | 0.76–0.92 | 0.000 | 0.916 | 0.63–1.32 | 0.641 |
| **Preterm gestation age at delivery (<37 weeks)** | **1.96** | **1.43–2.55** | **0.000** | **1.48** | **1.11–2.09** | **0.015** |
| **Post-term gestational age (≥ 42 weeks)** | **6.4** | **4.31–22.78** | **0.000** | **5.05** | **4.49–32.0** | **0.001** |
| **Term gestational age at delivery (37–41 weeks)** | **1** | | | **1** | | |
| Birth weight (kg) | 1.000 | 0.99–1.00 | 1.00 | 1.000 | 0.99–1.00 | 1.00 |
| Gender – female | 1.101 | 0.69–1.73 | 0.68 | 1.111 | 0.64–1.90 | 0.701 |
| Gender – male | 1 | | | 1 | | |
| | | HOSPITAL | | | FACTORS | |
| **Centre (Amana)** | **0.315** | **0.18–0.54** | **0.000** | **0.330** | **0.18–0.61** | **0.000** |
| **Centre (Mwananyamala)** | **2.971** | **1.23–7.19** | **0.016** | **`3.914** | **1.44–10.62** | **0.007** |
| **Centre (Temeke)** | **1** | | | **1** | | |

**Note:** Goodness of fit (Hosmer Lemeshow) $\chi^2$ test = 4.61 df = 4.

three hospital facilities are all in the same service rank (regional referral hospitals), and at present public referral facilities, managed and owned by the central government, we do believe the observed statistical findings to be due to (an)-other hidden factor(s) that is/are rampant but never documented/studied before. We thus, wish to call for more analytical studies on this specific research agenda.

Our current statistics on *early newborn deaths* and the associated factors are not unique to Dar es Salaam/Tanzania environment. In fact, previous reports from other parts of Africa have yielded near the same findings. For instance, Weddith and his colleagues in Nouakchott, Mauritania reported a point prevalence of early newborn deaths of around 34.7% [13]. It was a cross-sectional study on factors associated with early neonatal mortality published back in 2019 [13]. The estimates from Mauritania, though alarming were for all practical purposes not significantly different from ours [13]. Other investigators who reported findings on risks of early neonatal mortality from referral hospitals included Tesfaye and colleagues whose reported statistics was more than half (54.7%) of studied babies reportedly dying within the first seven days of life [14].

Clinical factors associated with early newborn deaths were also assessed in this study. On statistical grounds, respiratory rate and gestational age at birth were significantly observed to be associated with chances of early newborn deaths in our current linear model findings. Specifically, whereas tachypnoeic (faster than normal breathing) newborns had about 2.3 folds statistically significant increased risks of death up and above eupnoeic (normal breathing rate) newborns, those with bradypnoea (slower than normal breathing rates) had up to 2-folds statistically significant increased risks of death compared to eupnoeic newborns, all other factors in check. We have several alternative explanations for those findings. First, since the number of bradypnoeic children were relatively fewer, it is equally likely that bradypnoea was a spurious observation in view of the statistical artefacts due to smaller number of bradypnoeic (relative to tachypnoeic and eupnoeic) newborns. That is a design weakness that could not be corrected during analysis.

Otherwise, there are several reports that suggest tachypnea (and not bradypnoea) to be a more sensitive indicator of hypoxia among ill-neonates [15,16]. For instance, Rajesh and colleagues reported tachypnoea (breathing rate > 60 b/min) to be a more sensitive (sensitivity = 80%) indicator of hypoxia in critically ill newborns [15]. Thus, we could not rule out the possibility for the *lag factor* of bradypnoea in the risks of deaths in our cohort. Likewise, whereas preterm delivered newborns had up to 55% increased chances of death compared to term delivered newborns, those who were post-term deliveries had up to 5-folds increased risks of death compared to term delivered newborns.

Findings on clinical factors associated with early newborn deaths in this study are consistent with others reported (including in Tanzania) before [17–19]. For instance, Mangu and colleagues study on trends and patterns of newborn deaths in Tanzania revealed neonatal respiratory rate and distress to account for up to 20.8% of the causes for early newborn deaths [3]. Reyes and colleagues identified gestational age, respiratory rate, Apgar scores as significant risks associated with early neonatal deaths [18]. Moreover, whereas Reyes study findings were from outside Africa [18], Moshiro and colleagues [19] as well as Mangu and others [3] reported study findings were reported from Africa, in particular Tanzania.

Our current findings have several added advantages compared to others published before on the same topic. We conceived a prospective analytical study design and hence by design we could capture potential temporal association if present. Likewise, by virtue of our study design, both relative as well as absolute risks could be established since by default we could get population at-risk estimate. Most other scholars who reported similar findings had either cross-sectional [13,17] or retrospective observational designs [3,6,14]. Besides, the final analysis incorporated multivariable binary logistic regression model, and therefore potentially managed to account for common confounders during estimation of factors reported. However, there are a number of important limitations worth noted in this context. For instance, the entire study duration was limited by duration to six calendar months. There is palpable evidence in literature that shows mortality data in humans (neonates inclusive) to be affected by seasonality [20–23]. Thus, future studies on the same topic needs to account for the annual/seasonal variation of deaths among neonates, something that was beyond the current study power and design.

Besides, apart from the technical limitation in study duration coverage, the current study collected information from regional referral hospitals. Therefore, we could not completely rule-out potential referral biases in our findings. It is highly likely that the current estimate of early neonatal mortality to be an under-estimate of the true statistic, in lieu of the fact that we included only those newborns who were delivered at those facilities as well as considering the fact that normally referral bias results into negative biases, and hence likely under-estimation of the true statistic [24–26]. Moreover, it is important for readers and decision makers in mortality statistics to be aware of inherent limitations associated with our methods of analyzing current data. We relied on linear model analysis to derive our inferences. We thus wish to caution for and against possible *overfitting* and/or *mortality risks perceptions*.

The above two phenomena are common pitfalls in linear data analysis among both statisticians and demographers. To what extent do factors related to early neonatal deaths in real life obeying linear assumptions to be captured by our model is still thought provoking! However, on the basis of comparative crude and multivariable analysis done, our findings should be considered predictive enough on statistical grounds pending availability of better mechanistic model to explain better the interplay of those factors in real-world clinical scenario. Moreover, the fact that our linear model fitness was deemed statistically insignificant on Hosmer- Lemeshow Chi-square test, pinpointed to a probable linear model fitness. This gets an upper hand after realizing the fitted linear model had an intercept that was statistically significant associated with probable survival chances among studied babies. Thus, for all practical purposes, we do believe we managed to capture for factors associated with early neonatal mortality in the fitted linear model.

Otherwise, considering extensive global health efforts geared towards reduction of neonatal deaths, and especially early neonatal deaths throughout the world, and the marginal benefits vivid quantitatively, we have speculated possibilities of *latent factors* to be responsible for the *last mile delivery effect* against early neonatal survival chances. We have decided to speculate the 'last mile delivery effect' to be 'latent factors'-equivalent and coin it as *residual factors for early neonatal mortality*. We do believe the hypothesized '*residual factors for early neonatal mortality*' to have a potential for deleterious health consequences for newborns who managed to survive post-early neonatal era. We wish to treat that hypothesis as a potential research question worth future exploration. At present, there are mounting evidence that suggest transient and chronic ill-health in early childhood to be responsible to a myriad of negative health consequences throughout human lifespan [27–29]. We do hypothesise some of the '*residual factors*' to be answerable for chronic childhood and adult diseases, rampant in present day Africa, including Tanzania [30–37]. However, most clinical research studies on mortality statistics across the entire human lifespan are hampered by lack of reliable and validated tools [38–43]. We have started joining hands locally for a pioneering move towards a global solution for not only delaying deaths in neonatal era but also establishing the pioneering move to account and prevent early life precursors of deaths [44].

## Supporting information

**S1 Data. Minimal dataset.**
(XLS)

## Acknowledgments

We wish to convey our heartfelt gratitude to all newborn babies who participated in this study. Our sincere votes of thanks should also go to academic and administrative staff members of Hubert Kairuki Memorial University in Dar es Salaam, Tanzania. They provided a supportive academic environment that was necessary to make this study a success. Besides, special votes of thanks should go to staff at Amana, Mwananyamala and Temeke regional referral hospitals in Dar es Salaam, Tanzania. We also wish to convey our token of appreciation to Mr Faustine Mayunga, Drs Godfrey Swai and Mathew Mwanjali at Registered Trustees of Ultimate Family Healthcare in Dar es Salaam, Tanzania for their intellectual and moral support during data analysis and scientific writing workshops. Moreover, a sincere vote of thanks to

*the Coffeeshop Clinical Research & Scientific Writing* joint-programme between Registered Trustees of Ultimate Family Healthcare and I-Katch Technology Ltd in Dar es Salaam. The knowledge and skills acquired during the scientific study designs, data analysis and even scientific writing courses will be forever memorable in our lives!

## Author contributions

**Conceptualization:** Grace Frank Mhando, Salvatory Florence Kalabamu, Maulidi Rashidi Fataki, Christina Clinton Galabawa, Kelvin M. Leshabari.

**Data curation:** Grace Frank Mhando, Kelvin M. Leshabari.

**Formal analysis:** Grace Frank Mhando, Salvatory Florence Kalabamu, Maulidi Rashidi Fataki, Kelvin M. Leshabari.

**Funding acquisition:** Grace Frank Mhando.

**Investigation:** Grace Frank Mhando, Maulidi Rashidi Fataki, Kelvin M. Leshabari.

**Methodology:** Grace Frank Mhando, Salvatory Florence Kalabamu, Maulidi Rashidi Fataki, Christina Clinton Galabawa, Kelvin M. Leshabari.

**Project administration:** Grace Frank Mhando, Salvatory Florence Kalabamu.

**Resources:** Grace Frank Mhando.

**Software:** Grace Frank Mhando, Kelvin M. Leshabari.

**Supervision:** Salvatory Florence Kalabamu, Maulidi Rashidi Fataki, Kelvin M. Leshabari.

**Validation:** Salvatory Florence Kalabamu, Maulidi Rashidi Fataki, Christina Clinton Galabawa, Kelvin M. Leshabari.

**Visualization:** Grace Frank Mhando, Salvatory Florence Kalabamu, Maulidi Rashidi Fataki, Christina Clinton Galabawa, Kelvin M. Leshabari.

**Writing – original draft:** Grace Frank Mhando, Kelvin M. Leshabari.

**Writing – review & editing:** Grace Frank Mhando, Salvatory Florence Kalabamu, Maulidi Rashidi Fataki, Christina Clinton Galabawa, Kelvin M. Leshabari.

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
