## [Decision Letter · Decision Letter 0]

16 Mar 2025

Dear Dr. Leshabari,

Thank you for submitting your manuscript to PLOS ONE. After careful consideration, we feel that it has merit but does not fully meet PLOS ONE’s publication criteria as it currently stands. Therefore, we invite you to submit a revised version of the manuscript that addresses the points raised during the review process.

We look forward to receiving your revised manuscript.

Kind regards,

Chika Kingsley Onwuamah, Ph.D.

Academic Editor

PLOS ONE

Journal Requirements:

Reviewers' comments:

Reviewer's Responses to Questions

**Comments to the Author**

1. Is the manuscript technically sound, and do the data support the conclusions?

Reviewer #1: Partly

Reviewer #2: Partly

Reviewer #3: Yes

Reviewer #4: No

2. Has the statistical analysis been performed appropriately and rigorously?

Reviewer #1: No

Reviewer #2: No

Reviewer #3: Yes

Reviewer #4: No

3. Have the authors made all data underlying the findings in their manuscript fully available?

Reviewer #1: Yes

Reviewer #2: Yes

Reviewer #3: Yes

Reviewer #4: Yes

4. Is the manuscript presented in an intelligible fashion and written in standard English?

Reviewer #1: Yes

Reviewer #2: Yes

Reviewer #3: Yes

Reviewer #4: Yes

Reviewer #1: Thank you for the opportunity to review the manuscript “Predictors of Early Neonatal Mortality at Public Regional Referral Hospital in Dar es Salam Tanzania: A prospective observational hospital-based study.

General Overview: The proposal addresses important issues concerning child health in the public health space, specifically neonatal mortality and its associated predictors.

The Outline of the manuscript is in tandem with the journal.

Abstract:

Methods. The target population should be rephrased as all neonates in the hospital. The outcome is the death of neonates within 72 hours of life.

The phrase “Unless otherwise specified, an α-level of 5% was used as a limit of type 1 error rate in findings” should be deleted from the material and methods in the abstract as it is a hanging tense”.

Conclusion: The word relatively should be deleted from the sentence.

Introduction:

Well written, with the need to avoid some informal words in between tenses.

Rewrite/delete the tense, for instance, as: The three commonest factors associated with early neonatal mortality are…

Paragraph 2, line 3: Delete the word “Antagonistic”. Please kindly explain the sentence, “However, in some reported statistics, these factors are either contradictory or antagonistic, dependent on a number of mechanisms”

Methods:

What is the difference between the study population and the targeted population? I guess the target population is the author's choice when defining the study’s primary outcome.

The authors also need to clarify the participants enrolled. Were the participants only those delivered in the three referral hospitals, or did they include those referred from other hospitals for continued care in the three centres? This is important because there seems to be some conflicting tense of eligibility in the sample procedure sub-section and the recruitment and study duration sub-section.

The last sentence in the data collection procedure section appears redundant advise to delete it.

Variable Design and Measurement: If children delivered out of the referral facilities were involved in the study. Then, the place of delivery should be included as a variable.

Ethical clearance: Please do rephrase the tense: “that their babies will be subjected to minimal physical discomforts, especially during physical examinations as well as where to send their queries (if any) regarding the study/investigations in general” to the babies will be subjected to routine physical examination on enrollment

What is the formula for computing the Early Neonatal mortality rate

Results:

The results section is poorly written. It is just only a display of tables without phrases or tenses to describe the resulting output. The authors need to redo the result section significantly.

Example: Tabe 1:

The median age of participants enrolled in the study was 6 days, with a male-to-female ratio of (1:0.8). 71% of the participants were delivered via SVD, 26% via cesarean section

In table 1: Were the assisted delivery through SVD or C/S

Can the Place of delivery be better categorised? I presume all the hospital deliveries were delivered in the three primary referral facilities in the study, while the health centre, dispensary, home and others were out of the facilities. What does the other imply since it is only one subject?

Table 2: This table is confusing. The authors should help depict more clearly. The need for clear definitions of terms or diagnoses is important.

Where all the RDS associated with Prematurity or the MAS associated with Birth Aspyxia

Discussion:

Poorly written discussion.

The first paragraph appears more as postulations rather than the true summary of the findings from the results.

The assertion of residual factors seems unclear. What does the author mean by the term residual factors?

Reviewer #2: The manuscript needs major revisions in all the sections, especially the methods, result and discussion areas. These have been highlighted in the review report. A definition of terms used should be included to aid understanding of the subject.

Reviewer #3: The authors are invited to address the section on “Data collection procedure” as follow:

1.Data quality issues need attention. Clinical and laboratory data were collected on neonates from three hospitals hence three different research teams thus raising data quality assurance concerns. It is commendable that the instrument used to collect data was pilot-tested. However, there was no indication that the investigators were trained to use the tool. What did the authors do to address data consistency was assured? The study collected laboratory as well as clinical data. Did the Pediatric MDs (residents) attend all the deliveries and conducted the Apgar scoring or the midwives? Did all hospitals use the same (type, model) weighing scales and pulse oximeters for the babies? Did the authors use one lab or each hospital conducted their laboratory analysis (eg blood glucose)? Quality control measures for the clinical measurements and labs should be reflected in the write up.

2.Respiratory rate was one of the clinical parameters that predicted mortality as reported in this communication. In what way did the respiratory rate contribute to mortality? One would have expected that the pathological conditions that have bearing on the respiratory rate should have been factored into the analysis. Could this be reviewed and addressed?

Reviewer #4: The title sounds good with the intention of guiding Paediatricians and medical officers on the predictors of Neonatal mortality and ways to prevent such outcomes. However, the authors did not derive their conclusions from the study but a lot of assumptions. The authors are expected to respond to the following queries

1.What were the inclusion and Exclusion criteria

2 Three centres were chosen and the number of new born mortalities cannot be same therefore proportionate numbers should have been highlighted.

3. Were the children with congenital malformations included or excluded from the study

4. In the study " was premature rupture or prolonged rupture of membranes or both documented.

5.Results were not clear with too many jumbled information

6.How many babies actually died within the period was not stated except its readers teased out the number from table 1(370).Of these 370, from the table only 4 had their mothers less than 18 yrs and 35 > 35years. what about the others.

7.Are the causes of deaths of the same from all the centres

8. No table showed the predictors of neonatal mortality which is the title of this manuscript. Table 2 showed only the diagnosis and outcome but not predictors, neither did table 3. Some of the variables shown in table 3 were wt, sex , temp etc. What are the effects of these variables on neonatal mortality in terms of weight difference -prematurity,Large for age or small for age. How many female or males babies died, or what effect does hypothermia or hyperthermia or maternal age have on neonatal mortality.

8.Discussion was not based on the findings from the study eg such statement among others identified " its appears majority of the reported deaths occur during the first 24 hrs" . No evidence to suggest that from the results. Hence the discussion is not in line with study outcome or findings.

9.The use of the word "residual factors'. what are these residual factors. Did the study prove it. The authors will have to be cautious with use of such words in the manuscript.

The topic sounds good but the authors have not proven or highlighted the predictors of mortality in newborn babies in Tanzania such that a doctor practising in that part of the world would watch out for such predictors that can lead to mortality

**Do you want your identity to be public for this peer review?** For information about this choice, including consent withdrawal, please see our Privacy Policy

Reviewer #1: **Yes: ** Abideen Salako

Reviewer #2: No

Reviewer #3: No

Reviewer #4: No

---

## [Author Response · Author response to Decision Letter 1]

29 Apr 2025

REVIEWERS’ QUERIES/RECOMMENDATIONS SECTION

[PAGE NUMBER] AUTHORS’ RESPONSE CHANGES MADE

[YES/NO]

1. Abstract - Methods. The target population should be rephrased as all neonates in the hospital

Abstract

[pp. 2 ]

Indeed, in this clinical research, ‘target population’ was neonates who died within the first 72-hours of delivery.

All neonates in the hospitals were the study population and not the target population.

No

2. The phrase “Unless otherwise specified, an α-level of 5% was used as a limit of type 1 error rate in findings” should be deleted from the material and methods in the abstract as it is a hanging tense”.

Abstract

[pp. 2]

Deleted completely as suggested by the reviewers.

Yes

3. Conclusion: The word relatively should be deleted from the sentence

Abstract

[pp. 2]

Deleted completely as recommended by the reviewers

Yes

REVIEWERS’ QUERIES/RECOMMENDATIONS SECTION

[PAGE NUMBER] AUTHORS’ RESPONSE CHANGES MADE

[YES/NO]

4. The WHO definition of early neonatal death is death occurring in the first 7 days of life, not the first 3 days

Introduction

[pp.3] In this research paper, we adopted the American Academy of Pediatrics definition of ‘early neonatal death’ that refers to death occurring in the first 72-hours (3-days) of life.

We have also included ‘definition of terms’ in the manuscript

Yes

5. The “demographic data on neonatal death risk factors” are said to be “controversial”, but neither the factors nor the controversy are stated/explained

Introduction

[pp. 3]

We provided/stated the controversies associated with the term ‘early neonatal deaths’ in the introduction section.

Specifically, we highlighted the differences in the definition of ‘early neonatal death’ as to what appears to be informally referring to ‘American vs European’ standards. We specified that there appears to be confusion in literature as whereas American Academy of Pediatrics considers ‘early neonatal period’ to be within the first 72-hours after delivery, in Europe (including WHO) definition, the same observation refers to the first seven days of life

Yes

1. What is the difference between the study population and the targeted population? I guess the target population is the author's choice when defining the study’s primary outcome

Methods

[pp. 5]

Target population is defined in a standard format as population to which the primary research findings/outcomes will be applicable.

No

2. The authors also need to clarify the participants enrolled. Were the participants only those delivered in the three referral hospitals, or did they include those referred from other hospitals for continued care in the three centres? This is important because there seems to be some conflicting tense of eligibility in the sample procedure sub-section and the recruitment and study duration sub-section.

Methods

[pp. 5]

We only included newborns delivered at the study sites during this study period only. It was not clear where was the source of confusion to readers in the initial draft

Specifically, it is vividly written as

Sampling procedure: All newborns born at Amana, Mwananyamala and Temeke between 17 May and 16 November 2023 were eligible to be included into the study.

Recruitment and study duration: Newborns were recruited immediately after birth at the study sites. Recruitment was from 17th May 2023 to (and including) 16th November 2023.

No

1. The last sentence in the data collection procedure section appears redundant advise to delete it.

Methods

[pp. 6]

Deleted as recommended by reviewers

Yes

2. Variable Design and Measurement: If children delivered out of the referral facilities were involved in the study. Then, the place of delivery should be included as a variable.

Methods

[pp. 6]

We only included children born at the study sites in this study.

No

3. What data did those children admitted after 3 days of life contribute to the study outcome?

Methods

[pp. 5]

They will contribute in subsequent analysis (secondary data analysis) that will be published in the near future.

Yes

4. Data quality issues need attention. Clinical and laboratory data were collected on neonates from three hospitals hence three different research teams thus raising data quality assurance concerns.

It is commendable that the instrument used to collect data was pilot-tested. However, there was no indication that the investigators were trained to use the tool.

Methods

[pp. 6] Data quality assurance concerns were kept at near zero possibilities since

1.The tool was pilot tested

2.Research team members underwent a formal 1-day ‘good clinical and ethics research practice’ workshops done during the time of tools’ pilot testing.

3.Since all the three regional referral hospitals are financed by the same donor sources, they normally have the same lab and imaging tools. We also conducted a quick test-retest reliability as well as content validity analyses of all test results with no resultant statistically significant differences in findings during pilot test.

No

1. Very importantly, there was no definition of terms used in the paper. The definition/explanation of terms would have greatly facilitated understanding of the work

Methods

[pp. 5]

We have included a sub-section on ‘definition of terms’ in the methods section. For ease legibility, we decided to include it as the second sub-section, immediately after the study design section.

Yes

2. Ethical clearance: Please do rephrase the tense: “that their babies will be subjected to minimal physical discomforts, especially during physical examinations as well as where to send their queries (if any) regarding the study/investigations in general” to the babies will be subjected to routine physical examination on enrollment

What is the formula for computing the Early Neonatal mortality rate

Methods

[pp. 7]

We have made the changes as per the recommendations by reviewers

We have also included the definition and formula for computing early neonatal mortality rate used in this analysis.

Yes.

3. The number of babies recruited was not explicitly stated, and follow up days are not a part of recruitment

Results

[pp. 8] Number of babies recruited were explicitly stated in the manuscript using newborn-time of follow-up since the study was prospective by design. In classical epidemiology, studies adopting prospective design report total recruitment using person-time of follow-up rather than an absolute number characteristic in cross-sectional studies.

No

4. The results section is poorly written. It is just only a display of tables without phrases or tenses to describe the resulting output. The authors need to redo the result section significantly.

Results

[pp. 8-10] Results section has been extensively re-written as depicted in the new submitted manuscript file

Yes

1. It is also stated that the 3 main factors associated with early neonatal deaths, viz, neonatal, maternal and hospital system factors can be either contradictory or antagonistic without any explanation/examples of the contradictions/antagonism

Introduction

[pp. 3]

We have provided summary of the justifications behind both contradiction and antagonism.

Contradiction: the discrepancy in the definition of early neonatal period (death) between what appears to be ‘American vs European’ definitions.

Antagonism: differences in the causes of ‘early neonatal deaths’ between advanced economies vs. ‘low and middle income countries’

Yes

2. Babies were said to have been followed up for seven days or till death, whichever came first. Since the outcome of interest was mortality in the first 3 days, what was the significance of follow up beyond the first 3 days of life?

Methods

[pp. 5]

Investigators designed a research protocol on analyses for both deaths at day 3 and day 7 of life by design since even though the primary research question relied on ‘incident neonatal mortality and associated factors at day 3’ but there was a secondary research question that focused on analyzing ‘incident neonatal mortality on day 7’.

The original research paper will be on ‘early neonatal mortality and associated factors within 3 days of life’ and later on there will be another publication (secondary analysis!) for ‘incident neonatal mortality within 7 days of life’

No

---

## [Decision Letter · Decision Letter 1]

29 May 2025

Dear Dr. Leshabari,

Thank you for submitting your manuscript to PLOS ONE. After careful consideration, we feel that it has merit but does not fully meet PLOS ONE’s publication criteria as it currently stands. Therefore, we invite you to submit a revised version of the manuscript that addresses the points raised during the review process.

We look forward to receiving your revised manuscript.

Kind regards,

Chika Kingsley Onwuamah, Ph.D.

Academic Editor

PLOS ONE

Reviewers' comments:

Reviewer's Responses to Questions

**Comments to the Author**

Reviewer #1: All comments have been addressed

Reviewer #2: (No Response)

Reviewer #3: All comments have been addressed

Reviewer #4: (No Response)

2. Is the manuscript technically sound, and do the data support the conclusions?

Reviewer #1: Yes

Reviewer #2: Partly

Reviewer #3: Yes

Reviewer #4: Partly

3. Has the statistical analysis been performed appropriately and rigorously?

Reviewer #1: Yes

Reviewer #2: No

Reviewer #3: Yes

Reviewer #4: No

4. Have the authors made all data underlying the findings in their manuscript fully available?

Reviewer #1: Yes

Reviewer #2: Yes

Reviewer #3: Yes

Reviewer #4: No

5. Is the manuscript presented in an intelligible fashion and written in standard English?

Reviewer #1: Yes

Reviewer #2: Yes

Reviewer #3: Yes

Reviewer #4: Yes

Reviewer #1: The authors has provided needed answers and made correction to the intial review. The manuscript read better now

Reviewer #2: Although the issues raised in the previous review were responded to, a lot of the responses are inaccurate and highlight the need for the authors to endeavour to understand the subject matter of the research better. The detailed explanation is in the attached review report.

Reviewer #3: The data issue raised by me has been adequately addressed, thanks.

Reviewer #4: It is good to note that the authors have responded to some of the queries highlighted in the previous review. However, there are some unanswered questions to ensure this article reflects the topic which is Predictors of Neonatal deaths in 3 hospitals in Da res Salam .

1. Neonatal death mortality were used in several sections of the write up. The authors either sticks to Neonatal deaths or mortality and not death mortality

2. Some statements require references eg."Besides,

there is palpable evidence that Dar es Salaam city is among cities with highest neonatal deaths

in Tanzania"

3. The author would need to clarify some statements eg."Specifically,

neonates born (and admitted) at study sites within 37 days of life were recruited into the study."

I thought is about predictors of neonatal deaths occurring within 72 hrs of life

4. The authors did not discuss the possible hospital system factors that may be responsible for neonatal deaths which may serve as predictors or are these part of the residual factors.

5. The authors should highlight the possible residual factors responsible for neonatal deaths even if the study does not capture or what comprises the residual factors

6The predictors of death from this study were Birth Asphyxia, Prematurity and RDS. The authors did not extensively discuss their findings and reasons for the observations.

7 Does sepsis play any role in the mortality , the data did not display these findings apart from temp. recordings. What were the other risk factors for sepsis which could have worsened the mortality observed. These risk factors were mentioned as variables to be captured but were not analysed in the result section.

8 Table 2.. Factors should be changed to diagnoses. What was the total number of deaths even though they could have had multiple diagnosis.

8. Table 3 is deficient with information on the predictors, needs improvement. Maternal and Neonatal factors responsible or as predictors are not exhaustive

9. Authors should discuss their findings and give reasons or proffer reasons why the observations and relate it with other studies. Reviews in the discussions limited.

10 "Otherwise, we do believe the hypothesized ‘residual factors for early neonatal mortality’ to have a

potential for deleterious health consequences for newborns who managed to survive post-early"" What are these residual factors. The readers would like to know what makes up the residual factors and learn from it so as to provide better care for our babies.

**Do you want your identity to be public for this peer review?** For information about this choice, including consent withdrawal, please see our Privacy Policy

Reviewer #1: **Yes: ** Abideen Salako

Reviewer #2: **Yes: ** Agatha N. David

Reviewer #3: No

Reviewer #4: No

---

## [Author Response · Author response to Decision Letter 2]

15 Aug 2025

Neonatal death mortality were used in several sections of the write up. The authors either sticks to Neonatal deaths or mortality and not death mortality.

Title – [pp. 1]

Abstract – [pp. 2]

Introduction – [pp. 3-4]

Methods – [pp. 5-7]

Results – [pp. 8-10]

Discussion – [pp. 11-13]

-Authors wholeheartedly agreed to use the word ‘deaths’ instead of mortality for wider legibility of the intended message.

YES

Early neonatal death is defined by the American Academy of Paediatrics (AAP) as death in the first 72 hours of life

This is incorrect; the AAP defines early neonatal death as death in the first 7 days of life, in line with the WHO (Paediatrics 2016. 137(5): e20160551). This was reaffirmed in December 2024.

Introduction - [pp. 3]

Methods – [pp. 5]

Authors accepted the standard definition as per the reviewers’ suggestion and hence had to change not only the definition of the term but also re-run the analyses that incorporated the former definition (early mortality ≤ 3 days – 72 hours) to a newer one of 7-days duration

YES

Some statements require references eg. "Besides, there is palpable evidence that Dar es Salaam city is among cities with highest neonatal deaths

in Tanzania”.

Introduction – [pp. 3]

Missing reference that had to be added

YES

The controversies are said to stem from the inconsistency between American and European/WHO definitions.

This is debunked by the congruity of the two definitions as shown above. I believe they were misled by the disparities discussed in one of their referenced articles (Ref. 5), which disparities in neonatal mortality rate (NMR) across several European countries, comparing mortality rates and gestational-age associated mortality between the countries with the lowest NMR and the other countries. This should be understood in context and has nothing to do with the stated objective of this study.

In addition, the antagonism/contradiction among maternal, neonatal and health system factors in NMR is yet to be explained. These unexplored health system factors may actually be responsible for the so-called residual factors that they believe contributed to the high mortality rate.

Introduction – [pp. 3]

The entire sentences have been deleted after changing the definition of early neonatal period from 3 days (72-hours) to 7 days post-delivery.

YES

The author would need to clarify some statements eg."Specifically,

neonates born (and admitted) at study sites within 37 days of life were recruited into the study."

I thought is about predictors of neonatal deaths occurring within 72 hrs of life

Methods – [pp. 5]

Authors changed the definition of early neonatal deaths from 3 days (72-hours) to 7 days.

YES

Follow up days beyond 3 days of life is said to be for another publication.

It should have been clearly stated that what was being reported in this manuscript is the 3-day NMR of a cohort of newborns followed up from birth to 7 days of life.

An answer to the previous reviews queries

Authors changed the definition of the concept ‘early neonatal period’/’early neonatal death’ from earlier 72-hours (3-days) to the current 7-days duration. The changes di not only affected the definition of terms but also changed even the analyses made by incorporating newborns up to 7 days.

YES

they said they only recruited babies born in the study hospitals.

This is refuted by this statement in the methods section: “Besides, we also recruited neonates who were referred/transferred from other lower hospitals to the study settings provided that they were ≤ 3 days old.”

Methods – [pp. 5]

Authors deleted that sentence and hereby declare that to have been an erroneous statement that was earlier refuted during protocol design. The error was introduced by one reviewer from IRB who directed investigators for those babies to be also included.

However, investigators rejected that idea during subsequent protocol submission for IRB since it was to affect the calculation of ‘incident early neonatal mortality’ to the extent that it would have rendered the calculation of that index impossible on statistical grounds.

YES

“Number of babies recruited were explicitly stated in the manuscript”

Results – [pp. 8]

Authors added the total number of babies recruited

YES

The authors did not discuss the possible hospital system factors that may be responsible for neonatal deaths which may serve as predictors or are these part of the residual factors.

Discussion – [pp. 11]

Authors included ‘hospital/centre’ as one among potential HOSPITAL SYSTEM factors. Specifically, we separately analysed each hospital. Besides, we included each of the three hospitals separately in the multivariable binary logistic regression analysis whereby Temeke regional referral hospital was taken as a reference hospital under HOSPITAL SYSTEM factor (refer to table 3)

NO

The authors should highlight the possible residual factors responsible for neonatal deaths even if the study does not capture or what comprises the residual factors Introduction – [pp. 3] By default, the definition of residual factors included ‘latent variables…’ and by default could not be explicitly stated by investigators even though potentially statistically estimated.

We wish to request the reviewer to consider the term ‘residual factors’ as per the definition proposed in the introduction section of the manuscript.

NO

Table 2. Factors should be changed to diagnoses. Results [pp. 9] Changed as per the reviewers’ recommendations YES

"Otherwise, we do believe the hypothesized ‘residual factors for early neonatal mortality’ to have a

potential for deleterious health consequences for newborns who managed to survive post-early"" What are these residual factors. The readers would like to know what makes up the residual factors and learn from it so as to provide better care for our babies

Discussion [pp. 12] Authors kindly request the reviewer to refer to the proposed definition of the term ‘residual factors’ explicitly stated in the introduction section on page 3 of the manuscript document.

By default, we hypothesized that ‘residual factors’ to be de facto latent variables. Latent variables by their very nature

cannot be stated though statistically can be estimated.

NO

---

## [Decision Letter · Decision Letter 2]

27 Aug 2025

Dear Dr. Leshabari,

We look forward to receiving your revised manuscript.

Kind regards,

Chika Kingsley Onwuamah, Ph.D.

Academic Editor

PLOS ONE

Journal Requirements:

Reviewers' comments:

Reviewer's Responses to Questions

**Comments to the Author**

Reviewer #2: (No Response)

2. Is the manuscript technically sound, and do the data support the conclusions?

Reviewer #2: Partly

3. Has the statistical analysis been performed appropriately and rigorously?

Reviewer #2: No

4. Have the authors made all data underlying the findings in their manuscript fully available?

Reviewer #2: Yes

5. Is the manuscript presented in an intelligible fashion and written in standard English?

Reviewer #2: No

Reviewer #2: The issues raised in the previous reviews have not been satisfactorily addressed. Some vital information is missing from the manuscript such as number of live deliveries and the actual number of babies who died. The factors associated with mortality are not sufficiently elucidated; the tables are not sufficiently explicit. A lot of data said to have been collected with the CRF appeared not to have been included in the analysis, and these may have facilitated better comprehension of the study. A lot of allusion was made to "residual factors of neonatal mortality", but there appears to have been no efforts expended to discover what these factors might be.

A more detailed review is attached.

**Do you want your identity to be public for this peer review?** For information about this choice, including consent withdrawal, please see our Privacy Policy

Reviewer #2: No

---

## [Author Response · Author response to Decision Letter 3]

11 Sep 2025

1. How many of the 370 admitted babies actually died: from table 2, we have a total of 175 deaths and 299 living (giving a total of 474). We are told that this is because some babies had multiple diagnoses (in the presence of multiple diagnosis, what was the primary cause of death?). However, if we took the total of 474, then 175 deaths would be 36.9%, which would be in agreement with what is reported in some literature of about a third of admitted neonates dying

Authors' responses: - Authors have re-written table no. 2 to incorporate the comparator values. Thus, the total number of newborns who died was 104 and vivid in the sum total of each reported diagnosis in the death column. We hope we have removed the unnecessary confusion to reviewer’s/readers via incorporating the comparator rows in table no. 2

- Indeed, it was erroneous to summate the gross/total numbers of deaths in table no. 2 due to existence of multiple diagnoses in some studied neonates.

e.g. about 45% of neonates who died of RDS on day 1 were also reported to have birth asphyxia immediately after birth

2. How many live births occurred in the study hospitals during the 6 months of the study? This should serve as the denominator for the neonatal mortality rate in number/1000 live births.

Authors responses: - 370 live-newborns were delivered and followed-up at study settings during the study period.

Note:

a. We excluded by design all newborns admitted in the wards but delivered at lower facilities or home prior to admission.

b. We also excluded by design all stillbirth deliveries.

3. As stated in the previous reviews, it would have been easier to use the number of deaths over the 2,212 days of follow up to give the number of deaths /100 days of follow up of neonates admitted after birth

Authors' responses: - Noted.

That is near exactly what authors did in estimating incident early neonatal death rates. However, instead of using number of deaths/100 days of follow-up of neonates admitted after birth, we rather estimated the early neonatal death rates by using number of deaths that occurred among live-births delivered at the settings in 2212 days of follow-up over (divided by) total number of live-births delivered at the study settings in 2212 days of follow-up. Thereafter, the resultant estimate was multiplied by a factor of 1000 to get the estimated early neonatal death rates per 1000-live births

- Otherwise, authors did not understand the source of the 100-days of follow-up parameter in the computation of early neonatal deaths rates.

4. The tables are difficult to interpret; for table 2 for instance, what was the comparator for birth asphyxia that gave the values: ꭓ2 of 20.4, Df of 1, and p value of 0.000. Same difficulty for all the other diagnoses. Table 3 is even more difficult for me to interpret.

Authors' responses:

- Noted and changed as per the reviewer’s advise/recommendations.

Specifically, in each neonatal diagnosis, a comparator row has been added for ease observation and calculation of both observed and expected values in the chi-square statistics. Thus, even though there are no changes in the overall numerics/message, authors considered it logical to include comparator row.

- Likewise, authors have made substantial changes to table 3 in order to facilitate easy understanding to readers. Specifically,

a. For all categorical variables that incorporated dummies, a reference category has now been included in the table.

b. Besides, a description on why some variables in table 3 are without reference values (they were treated as continuous variables) has also been justified in text.

5. A lot of the parameters said to be collected with the CRF, including maternal education, time of registration for ANC, number of ANC visits, maternal conditions like chorioamnionitis and antepartum haemorrhage (omitted were maternal hypertension and diabetes), cadre of health providers at delivery, whether babies were singletons or multiples, time to admission to neonatal unit, and so on were not included in the analysis. The addition of these parameters would have enriched the study.

Authors' responses:

- Authors have included parameters that were initially left out in the results section. They are currently justified as text either before or as numerics in table 1 and either before as text or as numerics in table 3.

- Besides, the justifications behind why some factors described in table 1(baseline characteristics) still did not appear in the final analysis has been made in text for readers/reviewers.

i.e. We included only those (maternal, neonatal or hospital-based) factors that satisfied the assumptions statistically associated with linear model fitting. Thus, for a factor to be included into the final multivariable binary logistic regression model, it had to show evidence of its residuals to be normally distributed, homoscedastic with neither significant multi-collinearity, auto-correlation nor deviation from linearity assumptions. Besides, for inclusion into the final multivariable model, its p-value must have been < 0.2 at univariate/crude statistics.

6. From the discussion, it appears the major interest of the authors are the “residual factors of neonatal mortality”. These unnamed and unknown factors are speculations at best and should properly be mentioned at the end of the discussion (not the beginning) as possible causes of the “high”mortality.

Authors' responses: Agreed and removed everything about ‘residual factors of neonatal mortality’ from the start of the discussion section and shift them to the end of the discussion section as per reviewer’s recommendations/advice

7. I still maintain that it is not sufficient to mention “residual factors which are not known but can be statistically measured”. The autors should have explored deeper to discover the reason(s) for the discrepancies in deaths among the 3 study hospitals. The design of the study should have included possible health system factors such as: are the number of healthcare workers of different cadres adequate for the patient load, are there delays in patient admission, surgical procedures, are there patient fees that may hinder provision of care, are there adequate supply of materials at all times, are there unnecessary bottle-necks that hinder prompt patient care, etc.?

Authors' responses: -Agreed and some changes made. For instance, we included the comparator/reference hospital (for comparison) in the final analysis. However, some suggested changes were not considered since:

1. All the studied hospitals are in the same hierarchy in the Tanzania’s referral health system, and being managed/supervised by the same central ministry of health, and therefore their supplies are replicate equally resources at the study settings. i.e. Amana, Mwananyamala and Temeke hospitals have all the same number of child health specialists, the same hospital protocol as well as following the same national guidelines – e.g. in Tanzania, all children under five years of age (including neonates) receive free service on a de jure basis. Thus, exploring anything that is already guaranteed by laws (and order) would attract penalties from the ethics review (IRB) boards.

2. Some of the unnecessary ‘bottlenecks’ have been hypothesized and acknowledged by authors as study limitations.

e.g. there are potential indications that ‘early neonatal deaths’ in Dar es Salaam hospitals to have ‘seasonal variation’. The fact that our study was designed for 6-months only and made to be observational by design, means exact mechanisms about causal factors were far beyond the objectives and design of this observational study.

---

## [Decision Letter · Decision Letter 3]

29 Sep 2025

Dear Dr. Leshabari,

We look forward to receiving your revised manuscript.

Kind regards,

Chika Kingsley Onwuamah, Ph.D.

Academic Editor

PLOS ONE

Journal Requirements:

Reviewers' comments:

Reviewer's Responses to Questions

**Comments to the Author**

Reviewer #2: (No Response)

2. Is the manuscript technically sound, and do the data support the conclusions?

Reviewer #2: Partly

3. Has the statistical analysis been performed appropriately and rigorously?

Reviewer #2: No

4. Have the authors made all data underlying the findings in their manuscript fully available?

Reviewer #2: Yes

5. Is the manuscript presented in an intelligible fashion and written in standard English?

Reviewer #2: No

Reviewer #2: This manuscript has been reviewed severally and major issues identified in these reviews have not been addressed. The details are in the attached report.

**Do you want your identity to be public for this peer review?** For information about this choice, including consent withdrawal, please see our Privacy Policy

Reviewer #2: **Yes: ** Agatha N. David

---

## [Author Response · Author response to Decision Letter 4]

16 Nov 2025

Reviewers’ comments/queries/recommendations Section

[Page number] Authors’ justifications/responses to the changes/status made Changes made

[Yes/No]

1. The result of early neonatal mortality rate (NMR) of 281/1000 live births cannot be correct.

Results

[pp. 8 ] We have made changes as per the reviewers’ recommendations/suggestions. We deleted the incident mortality figures and computed the prevalence of mortality among sick admitted newborns. Specifically, the sentence now reads as

“Besides, we also found the prevalence of deaths among followed-up babies to be 28.1% at the study sites during the study period”

Yes

2. While table 2 has been corrected, Table 3 remains a challenge. In talking about associations, it is crucial to explain what is associated with what in order to provide clinical clarity.

Results

[ pp. 10 ] We have made changes as suggested/advised by the reviewer.

Specifically, we categorized respiratory rates into 3 distinct groups namely tachypnea (RR> 60 b/min), eupnoea (RR of 40 – 60 b/min) and bradypnoea (RR < 40 b/min) and reported their estimates separately in each category

Likewise, for gestational age at delivery, we also categorized into three different groups and report their estimates accordingly.

Yes

Reviewers’ comments/queries/recommendations Section

[Page number] Authors’ justifications/responses to the changes/status made Changes made

[Yes/No]

3. They explained away the non-use of maternal and health system information said to have been collected in the CRF by using fancy statistical terms but they should find a way to incorporate and analyze all that data. The addition of these parameters will enrich the study

Results

[pp. 9 ]

We fitted a newer binary multivariable logistic regression model that was considered PARSIMONIOUS enough (with LR-goodness of fit χ2 test score of 4.61 with 4 degrees of freedom) that was considered to be linear fit for our predictor variables. However, all efforts to include those factors were rendered futile since:

1. Some failed on linearity test (e.g. birth weight) – TOLERATED but proved to be FUTILE in its predictive power.

2. Some had significant multicollinearity with the fitted linear model (e.g. number of available neonatologists, presence/absence of emergency neonatal resuscitation equipments) – NON-TOLERABLE as no LINEAR model could be estimated with precision at any level of significance tested!

3. Others displayed significant autocorrelation (e.g. time of delivery, birth order of the baby) - NON-TOLERABLE as no fitted linear model.

4. And yet others displayed significant heteroscedasticity (e.g. history of maternal fever, chorioamnionitis and antepartum haemorrhage) with the outcome variable (neonatal death = 1) – NON-TOLERABLE

Thus, whereas TOLEARANCE could be granted for VIOLATION of linearity assumption (i.e. birthweight) the rest were considered A NON-NEGOTIABLE for inclusion into the linear model as for sure they were to render the entire analysis useless due to EXTREME VIOLATION of linear model assumptions. And even though we decided to tolerate violation of linearity in order to confer reviewer’s recommendations, any sensible reader would be able to see that in table 3 – results section, since the resultant p-value on the univariate analysis were a unit (p-value = 1.000) unlike a maximum of 0.2 that were justifiably a threshold for model inclusion into the multivariable analysis. Thus, predictive power of the variable “birthweight” was almost NIL for early neonatal mortality in the fitted linear model (see table 3 on pp. 10)!

Yes

---

## [Decision Letter · Decision Letter 4]

24 Nov 2025

Predictors of early newborn deaths at Dar es Salaam public regional referral hospitals: A prospective observational hospital-based study

PONE-D-25-05428R4

Dear Dr. Leshabari,

We’re pleased to inform you that your manuscript has been judged scientifically suitable for publication and will be formally accepted for publication once it meets all outstanding technical requirements.

Kind regards,

Chika Kingsley Onwuamah, Ph.D.

Academic Editor

PLOS ONE

Additional Editor Comments (optional):

Reviewers' comments:

Reviewer's Responses to Questions

**Comments to the Author**

Reviewer #2: All comments have been addressed

2. Is the manuscript technically sound, and do the data support the conclusions?

Reviewer #2: Yes

3. Has the statistical analysis been performed appropriately and rigorously?

Reviewer #2: Yes

4. Have the authors made all data underlying the findings in their manuscript fully available?

Reviewer #2: Yes

5. Is the manuscript presented in an intelligible fashion and written in standard English?

Reviewer #2: Yes

Reviewer #2: The authors have satisfactorily addressed the issues raised in the previous reviews and the manuscript is now recommended for acceptance for publication.

**Do you want your identity to be public for this peer review?** For information about this choice, including consent withdrawal, please see our Privacy Policy

Reviewer #2: No

---

## [Editor Report · Acceptance letter]

PONE-D-25-05428R4

PLOS ONE

Dear Dr. Leshabari,

I'm pleased to inform you that your manuscript has been deemed suitable for publication in PLOS ONE. Congratulations! Your manuscript is now being handed over to our production team.

Kind regards,

on behalf of

Dr. Chika Kingsley Onwuamah

Academic Editor

PLOS ONE